# Two-Stage Origin of K-Enrichment in Ultrapotassic Magmatism Simulated by Melting of Experimentally Metasomatized Mantle

**Michael W. Förster [1],\*, Stephan Buhre [2], Bo Xu [1,3], Dejan Prelević [2,4] , Regina Mertz-Kraus [2] and Stephen F. Foley [1]**

[1] ARC Centre of Excellence of Core to Crust Fluid Systems and Department of Earth and Planetary Sciences, Macquarie University, NSW 2109, Australia; xubo@outlook.com.cn (B.X.); stephen.foley@mq.edu.au (S.F.F.)

[2] Institute für Geowissenschaften, Johannes Gutenberg Universität, 55099 Mainz, Germany; buhre@uni-mainz.de (S.B.); prelevic@uni-mainz.de (D.P.); mertzre@uni-mainz.de (R.M.-K.)

[3] State Key Laboratory of Geological Processes and Mineral Resources, China University of Geosciences, Beijing 10083, China

[4] Faculty of Mining and Geology, University Belgrade, Đušina 7, 11000 Belgrade, Serbia

\* Correspondence: michael.forster@mq.edu.au

**Abstract:** The generation of strongly potassic melts in the mantle requires the presence of phlogopite in the melting assemblage, while isotopic and trace element analyses of ultrapotassic rocks frequently indicate the involvement of subducted crustal lithologies in the source. However, phlogopite-free experiments that focus on melting of sedimentary rocks and subsequent hybridization with mantle rocks at pressures of 1–3 GPa have not successfully produced melts with $K_2O$ >5 wt%–6 wt%, while ultrapotassic igneous rocks reach up to 12 wt% $K_2O$. Accordingly, a two-stage process that enriches $K_2O$ and increases K/Na in intermediary assemblages in the source prior to ultrapotassic magmatism seems likely. Here, we simulate this two-stage formation of ultrapotassic magmas using an experimental approach that involves re-melting of parts of an experimental product in a second experiment. In the first stage, reaction experiments containing layered sediment and dunite produced a modally metasomatized reaction zone at the border of a depleted peridotite. For the second-stage experiment, the metasomatized dunite was separated from the residue of the sedimentary rock and transferred to a smaller capsule, and melts were produced with 8 wt%–8.5 wt% $K_2O$ and K/Na of 6–7. This is the first time that extremely K-enriched ultrapotassic melts have been generated experimentally from sediments at low pressure applicable to a post-collisional setting.

**Keywords:** lamproites; high-pressure experiments; ultrapotassic; K-enrichment; subduction zones

## 1. Introduction

"Alkaline igneous rocks" is an umbrella term describing all igneous rocks that are generally enriched in the oxide species $Na_2O$ and $K_2O$ relative to sub-alkaline (tholeiitic) rocks at similar $SiO_2$ [1]. The alkaline rocks include sodic and potassic series, depending on the dominant oxide species. Potassic rocks are defined as comprising $K_2O > Na_2O$ in wt% and include compositional ranges from leucite-bearing basanites to K-enriched rock variants like leucitites, lamprophyres, orangeites, shoshonites, and lamproites [2]. Ultrapotassic (UP) lavas are alkaline igneous rocks with >3 wt% of both $K_2O$ and MgO, in which $K_2O$/ $Na_2O$ ratios exceed 2 [3]. They also show extreme trace element and isotopic enrichment. Potassium-rich magmatism is a common feature in the post-collisional stage of orogenic belts and exhibits compositions ranging from basic to silica-rich. Direct melting of the primitive mantle, which has a K/Na of ~0.1 [4], and thus contains ca. 10 times less $K_2O$ than $Na_2O$,

cannot produce potassic alkaline magmas with $K_2O > Na_2O$ [5]. Instead, $K_2O$ has to be enriched at source, and the isotopic compositions of most potassic alkaline magmas probably lie within the metasomatically overprinted lithospheric mantle.

Two models try to explain the enrichment of $K_2O$ and K/Na within Earth's mantle: (A) a two-stage metasome "vein + wall rock" melting model [6], and (B) a single-stage recycled sediment/crust melting model (e.g., [7]).

(A) The metasome melting model is a two-stage model that proposes that the UP magmas are produced by melting of a modally metasomatized mantle source that has been enriched in phlogopite and pyroxene in a first stage [8–11]. These phlogopite-rich "metasomes" may be pyroxenites or glimmerites, most likely formed when K-rich liquids react with peridotite to form layers and veins [12–14]. Experimental studies have successfully produced melts of UP composition from phlogopite-bearing and phlogopite-veined peridotites [15–18]. This explanation often does not specify the ultimate origin of the glimmeritic metasomes or the K-rich liquids that generated them.

(B) In contrast with the above model, other studies propose direct melting of subducted sediments [19] and/or continental crustal material [20], which mix with peridotite to form mélanges that melt to form UP magmas [7] in a single stage. Subducted crustal rocks have generally low solidus temperatures of >675 °C, which strongly depend on pressure and volatile contents [20–22]. Therefore, when the felsic crustal material is buried to mantle depths, it will preferentially melt and react with surrounding peridotite. As continental lithologies are usually K-rich compared with mantle rocks (K/Na 1–2), they can provide the K-enrichment for the potassic–ultrapotassic melts [23,24]. The concept of single-stage formation of UP lavas from recycled crustal components is grounded in the isotopic and geochemical compositions of post-collisional lavas, which are enriched in $^{87}Sr/^{86}Sr$ and show trace element patterns that are similar to the average composition of globally subducted sediment (GLOSS) [25]. Other studies have suggested other recycled crustal components, including blueschists [26], terrigenous siliciclastic sediments [9], and marly sediments [27].

However, recycled crustal components do not necessarily have to melt directly to produce UP melts in a single stage, but may produce an intermediate pyroxenitic component as in model (A). This is evident from olivines within post-collisional lavas that exhibit extreme Fo (94) and NiO > 0.5 wt%, which indicate a role for ultra-depleted peridotites [12,28]. To account for the crust-like geochemical composition of the post-collisional lavas, these pyroxenites must be phlogopite-rich and characterized by high $^{87}Sr/^{86}Sr$ isotopic and sediment-like trace element compositions. Given the widespread occurrence of post-collisional lavas, there also has to be a commonly occurring mechanism that produces pyroxenites with crust-like isotopic and trace element compositions within the lithospheric mantle. Furthermore, experiments on sediment and peridotite hybridization at pressures of 1–3 GPa, which are equivalent to depths of the sources of post-collisional ultrapotassic rocks [28,29], produce melts that are restricted to <6 wt% $K_2O$ on average [30,31], and indicate the importance of phlogopite pyroxenites as an intermediate product in the generation of UP melts. These phlogopite pyroxenites may not melt immediately following their formation and may reside within the lithospheric for several hundreds of millions of years before they are activated by lithospheric heating and re-juvenation.

In this study, we simulate the metasome melting model (A) for the formation of UP melts using a two-stage experimental approach in which parts of an experimental product are re-used in a second experiment (e.g., [32]). The two experimental stages are (1) heating of two-layer charges consisting of carbonate-bearing siliciclastic marine sediment and dunite, resulting in a phlogopite-rich pyroxenite metasomatic reaction layer; and (2) partial melting of the phlogopite-rich pyroxenite synthesized in the first experiment to generate UP melts with $K_2O > 6$ wt%.

## 2. Experimental Strategy

The two-stage experimental approach directly simulates the formation of UP magmas from a mantle metasomatized by melts of sediments. The use of natural materials enables us to compare trace element patterns of experimental melts with those found in natural K-rich lavas. The two-stage

approach also provides realistic mineral assemblages as starting material for the second stage melting experiments. The two-stage hypothesis is based on the following observations:

1. An increasing number of studies imply that fore-arc regions of long-lived volcanic arcs with abundant calk-alkaline volcanism may act as an important host for phlogopite-rich pyroxenites [33,34]. Trenchward migration of volcanism leads to extreme potassium and trace element enrichment in the erupted lavas, indicating that the fore-arc mantle is strongly metasomatized. Enrichment of the fore-arc probably takes place during subduction of sediments that melt at temperatures as low as 675 °C to produce Si-rich melts [21,22]. These Si-rich melts rise from the slab surface and immediately react with peridotite to produce phlogopite pyroxenites [23,35,36]. As phlogopite pyroxenites, as well as phlogopite peridotites, have solidi of ~1100–1200 °C [15,16,37,38], far above the melting temperatures of sedimentary rocks, the cold (as low as 675 °C) sediment melts within the fore-arc are completely consumed in this process.

2. Mantle xenoliths from arcs [39] as well as from various locations within the lithospheric mantle (e.g., [40–42]) demonstrate the presence of phlogopite enriched metasomes.

3. Trace element and isotope compositions of post-collisional volcanic rocks indicate thorough mixing of recycled crustal components in the source region (e.g., [43]).

4. Trace element signatures of olivine phenocrysts indicate the presence of pyroxenite in the source [12,29].

5. UP lavas have to be produced by the reaction of a melt of a low solidus vein and wall-rock peridotite with higher solidus, because multiple saturation points including Ol, Opx, Cpx, and Gt/Sp on various fluid contents and species as well as oxygen fugacities are absent under any conditions in liquidus experiments [13].

6. Direct melting of sediments and hybridization with peridotite at <3 GPa cannot generate melts with $K_2O$ mass fractions exceeding 6 wt% on average (e.g., [31]).

7. UP magmatism shows a prolonged activity of tens of millions of years' duration within a given region, indicating that the recycled component is able to reside for a long time within the mantle lithosphere (e.g., [44,45]).

## 3. Materials and Methods

### 3.1. Starting Materials

The sedimentary starting material was acquired from the International Ocean Discovery Project (IODP) site ODP 161–976 B 18 X3 105–106.5 (Supplementary Data Table S2) and is a hydrous, carbonate-bearing, siliciclastic marine sediment with <10% carbonate. This sample was selected to determine how local Mediterranean sediment is involved in the formation of UP lavas in the same region. Three aliquots of the sediment were analyzed for H (1.83 wt%), C (8.18 wt%), and N (0.35 wt%) in an automated vario EL cube elemental analyzer (Elementar, Langenselbold, Germany) [46].

A dunite (sample ZD11–53) from the Zedang ophiolite (south Tibet, China) containing olivine (>97%), spinel (~2%), and clinopyroxene (<1%) was used as the depleted peridotite (Supplementary Data Table S2). Both samples were powdered in an agate mortar.

### 3.2. Experimental and Analytical Techniques

Experiments were carried out using a piston cylinder apparatus at the University of Mainz, which produces exceptionally well-preserved glasses owing to rapid quench rates realized by extra cooling channels through the bomb plate close to the WC cores [47]. For the first-stage experiments (3 GPa and 800–1000 °C), the dunite was placed as a distinct layer on top of the sediment in a 30/70 ratio and sealed in 4 mm diameter capsules. Starting materials were filled into carbon capsules, where $fO_2$ lies below the C + CO + $CO_2$ equilibrium, and these were placed into sealed platinum capsules. The assemblies consisted of $Al_2O_3$ spacers, a graphite furnace, B-type thermocouple, and a sintered $CaF_2$ outer spacer. Thermobaric conditions correspond to the fore-arc setting of a subduction zone [48]. All

assemblies were pressurized first and heated at a rate of 50 °C/min. Pressure and temperature were kept constant for 1–14 days. All charges were quenched to temperatures below 500 °C within 8 s [47]. After the experiments, capsules were cut in half longitudinally; one half was embedded in epoxy and polished for characterization of the charges, and the other half was prepared for the second-stage melting experiment.

For the second-stage melting experiment, the completed first-stage experiment was dissected with a scalpel to separate the metasomatized dunite from the reacted sediment. The experiment at 900 °C was used for this purpose because the dunite contained the thickest phlogopite layer. The size of the cut out is highlighted by the blue box in Figure 1. During this mechanical extraction process, the sample material disintegrated into smaller pieces, which were sealed together in a capsule measuring 2 mm in diameter.

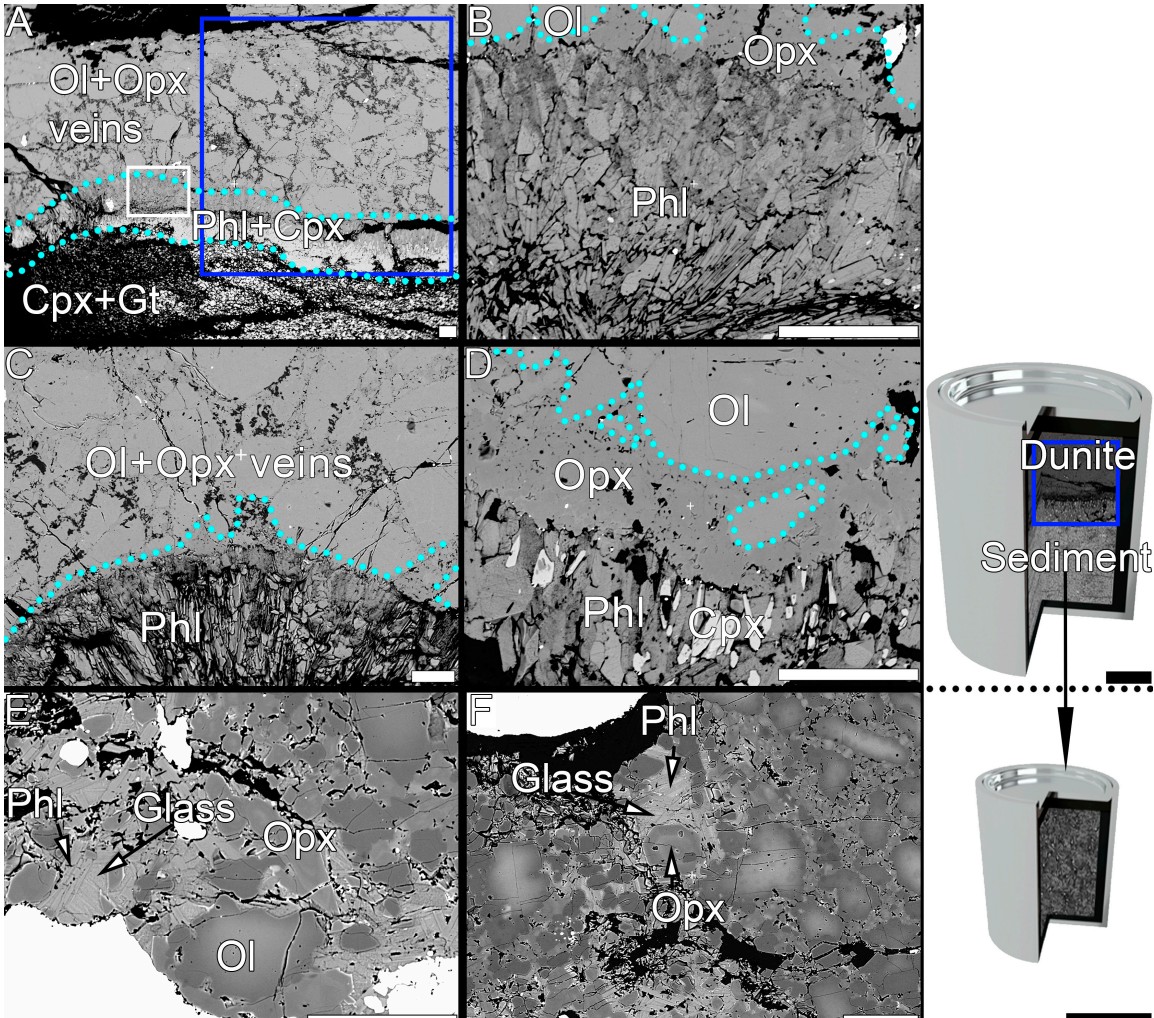

**Figure 1.** Backscattered electron images of reaction zones in high-pressure experiments. (**A**) Reaction experiment at 3 GPa/900 °C with dunite upper layer and sediment lower layer. Light blue dotted line highlights the extent of a 300–400 µm thick reaction zone composed of phlogopite and clinopyroxene. Blue square indicates the portion extracted for the second-stage experiment. (**B**–**D**) Close-ups of the upper part of the reaction zone (blue box in A). (**E**,**F**) Second-stage experiment in which the metasomatized dunite (blue square: reaction zone and dunite) of the experiment in (A) was melted at 2 GPa/1200 °C. All white scale bars (A–F) are 100 µm, while black scale bars for both capsules are 1 mm.

Major element contents of experimental run products were acquired using a JEOL JXA 8200 superprobe electron-probe microanalyzer (EPMA) equipped with five wavelength dispersive

spectrometers at the University of Mainz, Germany. All samples were measured using 15 kV accelerating voltage, spot sizes of 2 μm for silicate phases and 10 μm for glass phases, and a beam current of 12 nA with peak counting times of 20–30 s. A range of synthetic materials were used as reference materials. Analyses are presented in Supplementary Data Tables S1–S10.

Trace element mass fractions (Supplementary Data Tables S11–S14) were determined using laser ablation-inductively coupled plasma mass spectrometry (LA-ICP-MS) at the University of Mainz and solution ICP-MS at Macquarie University, Sydney.

Glass and mica from the 3 GPa/1000 °C experiment were ablated using an ESI NWR193 ArF Excimer laser ablation system (193 nm wavelength) equipped with a TwoVol2 ablation cell. The ablation rate was set to 10 Hz at 3 J/cm$^2$. The dry aerosol was transferred to an Agilent 7500ce mass spectrometer by a He–Ar mixed gas flow. Background signals were acquired for 20 s, followed by a dwell time of 30 s with spot sizes of 20 μm at each analyzed spot. For calibration, synthetic NIST SRM 612 was used with published values [49] and $^{29}$Si was selected as the internal standard using the SiO$_2$ mass fractions determined by EPMA. USGS BCR-2G was analyzed as an unknown in each run for quality control; measured values were within 10% of the data tabulated in the GeoReM database [50]. Data processing and reduction were carried out using the commercial software GLITTER 4.4.1 [51]. Trace-element mass fractions in the sediment were measured by solution ICP-MS. For quality control, USGS BHVO-1 and BIR-1 were measured as unknowns and values are within 10% of the data tabulated in the GeoReM database [50].

## 4. Results

### 4.1. First-Stage Experiments

Four first-stage reaction experiments were performed at 3 GPa at temperatures ranging from 800 to 1000 °C (Table 1). The breakdown of carbonate, clay, and organic material produces a mixed H$_2$O ± H$_2$ ± CO$_2$ ± CO ± CH$_4$ fluid, whose speciation strongly depends on pressure, temperature, and oxygen fugacity. Given carbon saturation, the decomposition of the sediment follows the following reaction: CH$_4$ + H$_2$O = 3H$_2$ + CO [52].

The sedimentary starting material contained 2.57 wt% K$_2$O with a K/Na ratio of 1.85 (Supplementary Data Table S2, K$_2$O/Na$_2$O = 1.65). All experiments show a reaction zone (80–400 μm) sandwiched between the sediment and dunite layers, which consists of a phlogopite and clinopyroxene corona on the sediment side as well as massive orthopyroxene on the dunite side of the reaction zone (Figure 1A–D). This reaction zone contains mainly phlogopite (~40 vol%–80 vol%) and clinopyroxene (~20 vol%–50 vol%), as well as minor amounts of orthopyroxene (<10 vol%) and garnet (<5 vol%), while accessory minerals include apatite and Fe–Ni-sulphides. Experiments at 800, 850, and 900 °C also contain veins of orthopyroxene with minor amounts of magnesite extending into the dunite layer from the reaction zone along olivine grain boundaries. The former sedimentary rock fraction contains ~15 vol%–22 vol% Mn-rich garnet (gr 0.31 alm 0.27 pyr 0.22 sp 0.19), ~20 vol%–50 vol% Na–Al-rich clinopyroxene (di 0.66 jd 0.34), ~20 vol%–40 vol% and glass in all experiments, while phengite at 850 °C and coesite at 800 °C were also found [53] (Tables 2 and 3). All first-stage reaction experiments produced hydrous silica-rich glasses, which were the result of 20%–40% melting of the sediment (Table 2, Supplementary Data Table S8), ranging from 70 wt%–79 wt% SiO$_2$ (Figures 2A and 3A) at 800–900 °C to 53 wt%–58 wt% SiO$_2$ at 1000 °C, when recalculated to a 100 wt% anhydrous composition. All glasses have <6 wt% K$_2$O, <16 wt% Al$_2$O$_3$, <4 wt% CaO, and <2 wt% Na$_2$O (Figure 3A–D), while K/Na ranges from 2.8 to 6.4 (Figure 4A). Clinopyroxene shows a continuous range of compositions across the reaction zone with increasing Na$_2$O (0.6 wt%–4 wt%) and Al$_2$O$_3$ (2 wt%–10 wt%) towards the sediment, while Na$_2$O decreases with temperature from 3 wt% to 1 wt% (Figure 2B).

**Table 1.** Melting phase relations of reaction experiments and second-stage melting experiment.

| # | Composition | T (°C) | P (GPa) | Melt (%) | Phases in Sediment Layer | Reaction Zone Phases | Phases in Dunite | Duration |
|---|---|---|---|---|---|---|---|---|
| 1 | Sediment/Dunite reaction | 800 | 3 | ~10 | Gt + Cpx + Phe + Ap + Cc + Coe + Si-rich melt | Cpx + Phl + Opx | Ol + Opx + Mgs | 7 d |
| 2 | Sediment/Dunite reaction | 850 | 3 | ~20 | Gt + Cpx + Phe + Ap + Si-rich melt | Phl + CPx + Opx + Ap + Fe–Ni Sulphides | Ol + Opx + Mgs | 13 d |
| 3 | Sediment/Dunite reaction | 900 | 3 | ~20 | Gt + Cpx + Si-rich melt | Phl + Cpx + OPx+ Ap + Fe–Ni Sulphides | Ol + Opx + Mgs | 14 d |
| 4 | Sediment/Dunite reaction | 1000 | 3 | ~30 | Gt + Si-rich melt | Phl + CPx + Opx | Ol + Opx | 4 d |
| 5 | Metasome melting (Reaction zone of experiment 3) | 1200 | 2 | ~20 | - | Ol + Opx + Phl + melt | - | 6 h |

Mineral abbreviations: Gt—Garnet, Cpx—Clinopyroxene, Opx—Orthopyroxene, Phe—Phengite, Phl—Phlogopite, Ap—Apatite, Cc—Calcite, Coe—Coesite, Fe–Ni sulphides—Iron–Nickel sulphides, Si-rich melt—Silicate melt >55 wt% $SiO_2$.

**Table 2.** Starting material and melt compositions.

| Sample | K/Na | $Na_2O$ | $K_2O$ | MnO | $SiO_2$ | MgO | FeO | $Al_2O_3$ | CaO | $TiO_2$ | Total |
|---|---|---|---|---|---|---|---|---|---|---|---|
| Sediment bulk | 1.57 | 1.78(6) | 2.46(9) | 2.17(9) | 54.9(4) | 3.93(6) | 3.6(2) | 17.8(1) | 13.5(2) | 0.82(3) | 100 |
| Dunite bulk | - | 0.03 | 0.00 | 0.13 | 40.44 | 49.08 | 10.06 | 0.08 | 13.50 | 0.01 | 100 |
| 3 GPa/800 °C Sediment/Dunite reaction melt | 2.9(6) | 1.6(4) | 3.9(4) | 0.13(4) | 75(1) | 0.4(4) | 0.4(1) | 15.3(4) | 2.5(6) | 0.25(2) | 89(1) * |
| 3 GPa/850 °C Sediment/Dunite reaction melt | 6(2) | 1.0(3) | 5(1) | 0.03(4) | 76(1) | 0.12(7) | 0.15(5) | 14.6(8) | 1.6(5) | 0.25(5) | 88(2) * |
| 3 GPa/900 °C Sediment/Dunite reaction melt | 6(2) | 1.1(2) | 6.2(8) | 0.13(3) | 74.6(4) | 0.50(8) | 0.88(4) | 12.9(5) | 3.3(5) | 0.33(2) | 82(1) * |
| 3 GPa/1000 °C Sediment/Dunite reaction melt | 4.0(5) | 1.3(2) | 4.5(1) | 0.9(1) | 56(1) | 5.5(3) | 3.2(2) | 16.1(2) | 11.3(7) | 1.03(7) | 81(1) * |
| 2 GPa/1200 °C second-stage melting | 7.1(6) | 1.3(1) | 8.4(2) | 0.37(8) | 58(1) | 3(1) | 1.0(4) | 17.8(9) | 9(1) | 0.63(7) | 88(2) * |

* Analytical totals, oxide compositions of glass measurements are normalized to 100 wt%. The number in parentheses represents one standard deviation on the last digit.

**Table 3.** Mineral compositions.

| Olivine | Na$_2$O | K$_2$O | MnO | SiO$_2$ | MgO | FeO | Al$_2$O$_3$ | CaO | TiO$_2$ | Total |
|---|---|---|---|---|---|---|---|---|---|---|
| 3 GPa/900 °C Sediment/Dunite reaction | 0.04(5) | 0.01(2) | 0.13(2) | 40.7(4) | 52.3(3) | 8.5(2) | 0.01(1) | 0.06(3) | 0.01(1) | 101(1) |
| 3 GPa/1000 °C Sediment/Dunite reaction | 0.02(1) | 0.01(1) | 0.20(6) | 39.8(2) | 50.2(1) | 8.3(2) | 0.02(2) | 0.04(1) | 0.02(1) | 98.6(2) |
| 2 GPa/1200 °C second-stage melting | 0.002(3) | 0.01(1) | 0.06(1) | 41.6(2) | 54.6(4) | 1.97(6) | 0.01(1) | 0.33(3) | 0.01(1) | 98.7(4) |
| **Phlogopite** | **Na$_2$O** | **K$_2$O** | **MnO** | **SiO$_2$** | **MgO** | **FeO** | **Al$_2$O$_3$** | **CaO** | **TiO$_2$** | **Total** |
| 3 GPa/800 °C Sediment/Dunite reaction | 0.30(7) | 8.40(9) | 0.10(3) | 43.6(7) | 26.0(5) | 1.13(9) | 12.7(4) | 0.6(5) | 0.59(3) | 95.0(9) |
| 3 GPa/850 °C Sediment/Dunite reaction | 0.15(4) | 9(1) | 0.04(3) | 43(1) | 24(2) | 3.7(5) | 13(1) | 0.2(3) | 0.7(2) | 94(1) |
| 3 GPa/900 °C Sediment/Dunite reaction | 0.5(1) | 9.5(2) | 0.10(3) | 41.5(6) | 24.1(5) | 3.7(1) | 14.1(4) | 0.2(4) | 1.4(1) | 95(1) |
| 2 GPa/1200 °C second-stage melting | 0.2(1) | 10.4(7) | 0.09(5) | 42(2) | 24(4) | 1.0(2) | 14.7(5) | 1(1) | 1.4(2) | 96(2) |
| **Phengite** | **Na$_2$O** | **K$_2$O** | **MnO** | **SiO$_2$** | **MgO** | **FeO** | **Al$_2$O$_3$** | **CaO** | **TiO$_2$** | **Total** |
| 3 GPa/800 °C Sediment/Dunite reaction | 0.45(8) | 10.4(2) | 0.06(4) | 49.2(8) | 3.6(1) | 1.0(1) | 28.2(5) | 0.15(8) | 0.9(1) | 94.6(7) |
| 3 GPa/850 °C Sediment/Dunite reaction | 0.18(2) | 9.8(6) | 0.04(3) | 49.7(7) | 5.6(3) | 0.6(2) | 27.0(3) | 0.09(2) | 1.0(1) | 94.9(8) |
| **Clinopyroxene** | **Na$_2$O** | **K$_2$O** | **MnO** | **SiO$_2$** | **MgO** | **FeO** | **Al$_2$O$_3$** | **CaO** | **TiO$_2$** | **Total** |
| 3 GPa/800 °C Sediment/Dunite reaction | 3.0(6) | 0.2(3) | 0.31(6) | 53(3) | 13(3) | 1.9(6) | 8(2) | 16(1) | 0.23(7) | 96(4) |
| 3 GPa/900 °C Sediment/Dunite reaction | 1.4(4) | 0.1(1) | 0.4(2) | 53.8(8) | 16(1) | 2.3(4) | 5(2) | 21(1) | 0.21(9) | 100.3(5) |
| 3 GPa/1000 °C Sediment/Dunite reaction | 0.84(8) | 0.03(4) | 0.71(6) | 51.9(6) | 14.9(7) | 3.1(3) | 5(1) | 21.9(2) | 0.22(6) | 98.6(4) |
| **Orthopyroxene** | **Na$_2$O** | **K$_2$O** | **MnO** | **SiO$_2$** | **MgO** | **FeO** | **Al$_2$O$_3$** | **CaO** | **TiO$_2$** | **Total** |
| 3 GPa/900 °C Sediment/Dunite reaction | 0.03(4) | 0.4(9) | 0.30(7) | 57(1) | 36.7(6) | 5.1(4) | 1(1) | 0.15(2) | 0.08(4) | 100.9(4) |
| 3 GPa/1000 °C Sediment/Dunite reaction | 0.02(2) | 0.03(4) | 0.43(4) | 54(1) | 34.1(9) | 4.9(2) | 2.5(6) | 0.7(3) | 0.08(3) | 98(2) |
| 2 GPa/1200 °C second-stage melting | 0.16(4) | 0.01(1) | 0.12(4) | 54.5(6) | 35.1(7) | 1.41(8) | 5.8(7) | 1.3(4) | 0.21(6) | 99.8(4) |
| **Garnet** | **Na$_2$O** | **K$_2$O** | **MnO** | **SiO$_2$** | **MgO** | **FeO** | **Al$_2$O$_3$** | **CaO** | **TiO$_2$** | **Total** |
| 3 GPa/800 °C Sediment/Dunite reaction | 0.2(1) | 0.2(3) | 7(2) | 39(4) | 5.9(5) | 12(1) | 21.3(6) | 11.7(8) | 0.9(2) | 99.4(1) |
| 3 GPa/900 °C Sediment/Dunite reaction | 0.07(4) | 0.02(2) | 5(2) | 40.7(8) | 13(3) | 11(4) | 22.8(6) | 9(2) | 0.5(3) | 101.3(7) |
| 3 GPa/1000 °C Sediment/Dunite reaction | 0.02(4) | 0.03(2) | 4.2(4) | 40.4(6) | 12(1) | 8.7(6) | 21.9(5) | 11.1(5) | 0.34(5) | 99.1(2) |

The number in parentheses represents one standard deviation on the last digit.

Trace element compositions of melts derived from the sediment were acquired from the 3 GPa/1000 °C experiment, which contained large pools of melt that were easily measurable by LA-ICP-MS. This melt is strongly enriched in large ion lithophile (LILE), high field strength (HFSE), and rare earth elements (REE), and shows negative anomalies at Nb, P, and Ti, which are inherited from the bulk sediment composition (Figure 5).

## 4.2. Second-Stage Experiments

The second-stage melting experiment was conducted on the phlogopite pyroxenite from the first stage experiment at 3 GPa/900 °C (Figure 1A, Supplementary Data Table S2), which showed the thickest metasome layer of all reaction experiments. As the phlogopite pyroxenite disintegrated upon transfer to the new capsule, the second-stage experiment was not layered (Figure 1E,F). The second-stage experiment was performed at 2 GPa/1200 °C, thermobaric conditions at which phlogopite pyroxenites are known to be partially molten [16]. At these conditions, a mixture of phlogopite, clinopyroxene, and orthopyroxene breaks down to form olivine and melt [16]. After the second-stage melting, the glass contains ~57.7 wt% $SiO_2$, ~8.4 wt% $K_2O$, ~17.8 wt% $Al_2O_3$, 9.36 wt% $CaO$, and 1.33 wt% $Na_2O$ (Table 2, Figure 3A–D), with K/Na ~7 (Figure 4A). Olivine appears zoned with lower FeO rims, indicating Fe-loss due to contact of the melt with the Pt-capsule. However, $K_D$ (Fe/Mg)$_{Ol}$/(Fe/Mg)$_{liq}$ attain 0.14 at ~10 wt% $Na_2O$ + $K_2O$, which is in accordance with $K_{D(ol/liq)}$ decreasing with alkali content of the liquid [54]. The glass-composition was calculated to result from 20% batch melting of the metasome containing 21 wt% phlogopite, 71 wt% orthopyroxene, and 8 wt% clinopyroxene (Supplementary Data Tables S2 and S9).

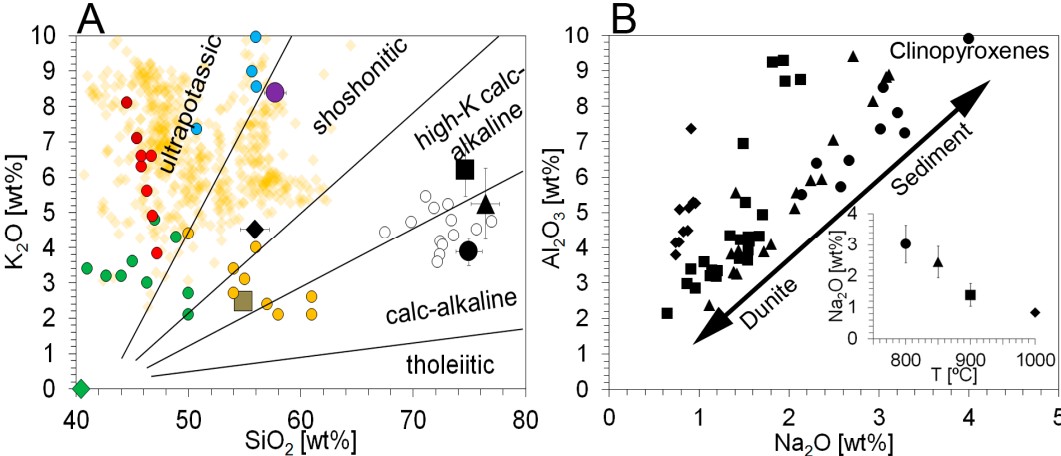

**Figure 2.** Melt (**A**) and clinopyroxene (**B**) compositions from reaction experiments (black circle 800 °C; triangle 850 °C; square 900 °C; diamond 1000 °C) and second-stage melts (purple circle) of this study, compared with the sedimentary rock starting material (brown square), and published data: melts of phlogopite harzburgite and lherzolite (dark red and red circles [15]), hybridized rhyolite–peridotite (green circles [18]), glimmerite–harzburgite (blue circles [16]), hybridized phyllite–dunite (open white circles [20,31]), and phlogopite–amphibole peridotite (orange circles [55]). Natural post-collisional lavas with K/Na >2 (yellow diamonds [9]) are shown for comparison. Melts from reaction experiments at 3 GPa/800–1000 °C are $SiO_2$-rich and have moderately high $K_2O$. In contrast, second-stage experiment melts (purple dot) have higher $K_2O$ and lower $SiO_2$, similar to natural ultrapotassic (UP) magmas. (B) Clinopyroxenes from reaction experiments show a range of compositions across the reaction zone, with higher $Na_2O$ and $Al_2O_3$ towards the former sedimentary rock. Symbols correspond to the experimental temperatures as in (A).

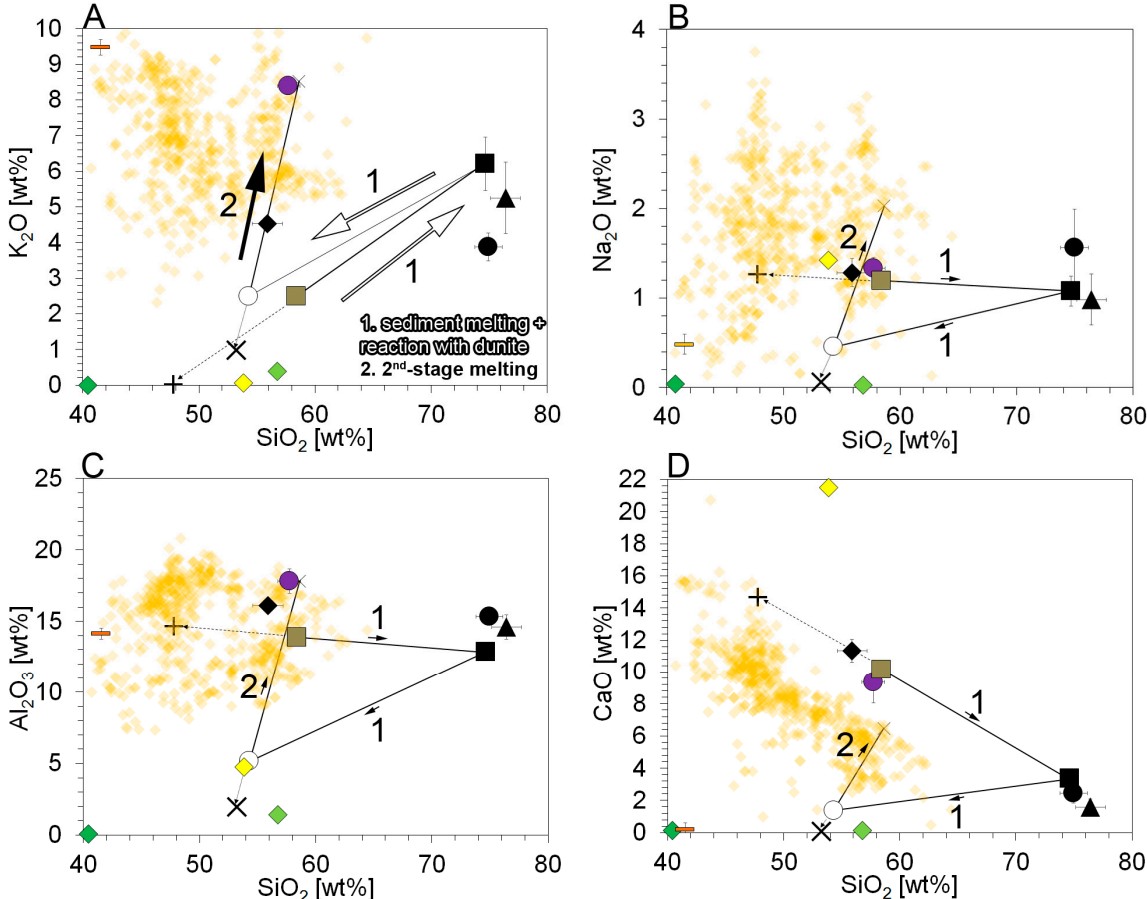

**Figure 3.** $K_2O$ (**A**), $Na_2O$ (**B**), $Al_2O_3$ (**C**), and CaO (**D**) versus $SiO_2$ showing the melt evolution of the 3 GPa/900 °C reaction and 2 GPa/1200 °C second-stage experiment as calculated by mass balance. In the first stage (white arrows), 40% melting of the sediment (brown square) generates an Si-rich melt (black square), leaving a residuum (black plus) comprising 40% garnet and 60% clinopyroxene (Supplementary Figure S1), a Ca-rich and K-poor composition. The Si-rich melt (black square) immediately reacts with olivine (dark green diamond), metasomatizing the adjacent dunite to phlogopite-pyroxenite (open circle). In the second stage, the metasomatized dunite melts to 20% crystallizing a residuum (black cross) comprising 16 wt% olivine, 56 wt% orthopyroxene, and 5 wt% phlogopite. Note that the calculated second-stage melt composition is close to the measured values (purple circle) for $K_2O$, $Al_2O_3$, and $SiO_2$ (A,C). However, $Na_2O$ and CaO are over- and underestimated, respectively, probably owing to contamination by residual sediment during transfer of the metasomatized dunite to the second-stage experiment. Other symbols: melts of sediment (black circle 800 °C; triangle 850 °C; square 900 °C; diamond 1000 °C), metasome (black star), reaction zone phases (phlogopite, orange rectangle; orthopyroxene, lime diamond; clinopyroxene, yellow diamond).

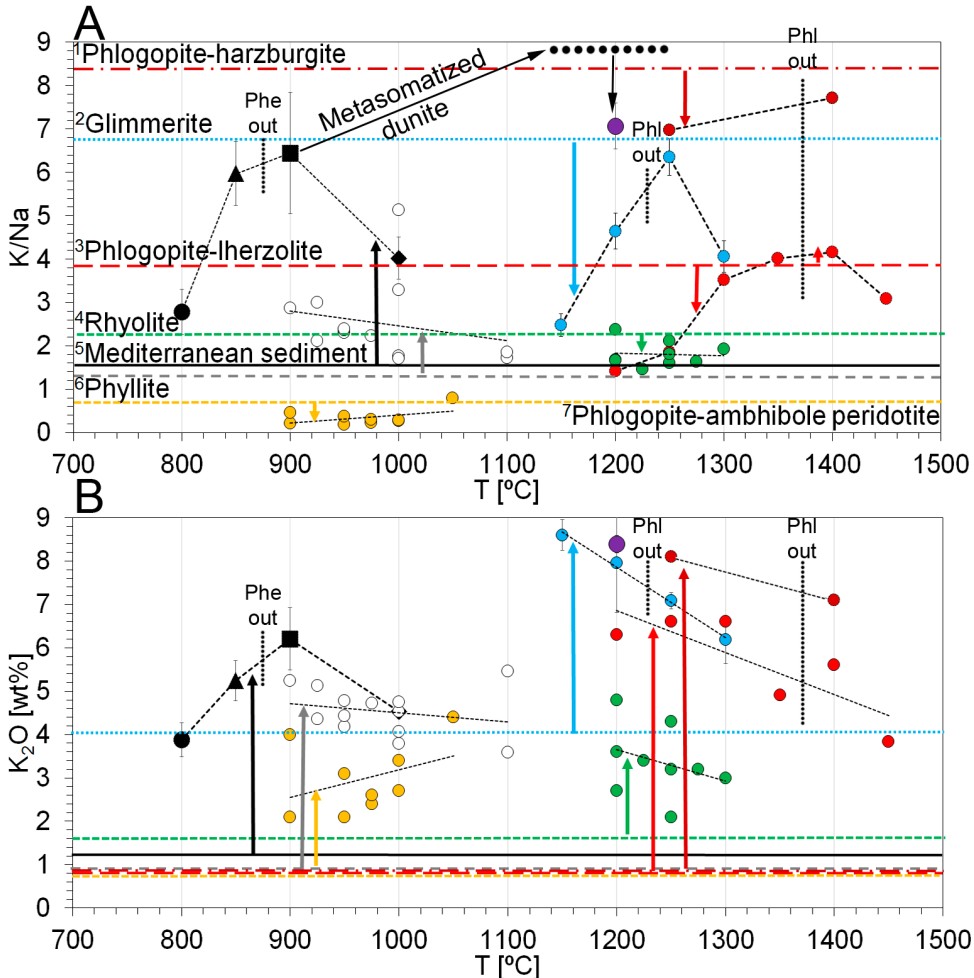

**Figure 4.** K/Na (**A**) and K$_2$O mass fractions (**B**) of melts versus experimental run temperature (black circle 800 °C; triangle 850 °C; square 900 °C; diamond 1000 °C) and second-stage melts (purple circle) of this study compared with published data for phlogopite harzburgite and lherzolite (dark red and red circles [15]), hybridized rhyolite/peridotite (green circles [19]), glimmerite/harzburgite (blue circles [16]), hybridized phyllite/dunite (open white circles [20,31]), and phlogopite–amphibole peridotite (orange circles [55]). (**A**) Horizontal lines represent starting materials and are color-matched to the symbols for the same experiments. K/Na in melts shows peaks related to phengite breakdown at ca. 900 °C and phlogopite breakdown at >1100 °C. Only experiments at <1100 °C generate melts in which K/Na ratios exceed those of the starting materials (upward facing arrows). At >1100 °C, the highest K/Na ratios approach those of the starting material (downward facing arrows). The arrow labelled "Metasomatized dunite" connects the 900 °C reaction experiment and the calculated starting composition (thick dotted line: Supplementary Data Table S2) of the second-stage experiment at 1200 °C. (**B**) K$_2$O mass fractions of experimental charges generally decrease with increasing temperature. Phengite-buffered K$_2$O mass fractions in siliceous melts first increase with temperature and then decrease after phengite breakdown. Maximum mass fractions of K$_2$O are <6 wt% for silica-rich melts (black symbols, white and green circles), while those buffered by phlogopite reach <9 wt% K$_2$O (blue, purple, and red circles).

## 5. Discussion

The generation of UP melts with 8.2 wt%–8.7 wt% K$_2$O, representing a >4 fold enrichment with respect to the original sediment source, is demonstrated unequivocally in our second-stage experiment. This is the first study that has generated UP melts with K$_2$O >6 wt% from recycled crustal lithologies with K$_2$O contents as low as 2.46 wt%, thus confirming the applicability of this two-stage process of K-enrichment. In the following discussion, we first compare our study with previous experiments in

similar systems and then discuss the application of our experiments to post-collisional lavas such as those occurring in the Alpine–Himalayan belt.

*5.1. Comparison with Previous Experimental Studies*

The recycling of crustal rocks into the mantle is a normal consequence of subduction and collision in the global tectonic cycle, constantly juxtaposing $SiO_2$-oversaturated crustal rocks with $SiO_2$-deficient mantle rocks. Isotopic and geochemical compositions of orogenic lavas in the Alpine–Himalaya orogenic belt (AHOB) undoubtedly document the effective recycling of crustal rocks to mantle depths [10,29,43,56,57]. The role of crustal recycling in the generation of UP post-collisional lavas differs between the two competing models presented above, with major uncertainties in terms of the process of hybridization and melt generation as follows:

(A)    UP lavas form by melting of phlogopite pyroxenites and phlogopite peridotites that are abundant within the continental lithospheric mantle [58,59]; this has been successfully simulated by experiments with UP melts (Figure 2), $K_2O$ 6 wt%–12 wt%, MgO >3 wt%, and low $SiO_2$ [15,16,60]. Infiltrating melts derived from sediments form phlogopite-bearing pyroxenites at the expense of peridotites following the reaction melt + olivine → phlogopite + pyroxene [23,35,61]. This process is concentrated in the fore-arc, where hydrous sediments melt and infiltrate the mantle wedge because of their low solidus temperatures and high volatile contents [34].

(B)    UP lavas form by direct melting and hybridization of crustal rocks with mantle peridotites in a single stage process at mantle depths [7,62]. Melt inclusions in garnets in ultrahigh-pressure paragneiss (5 GPa) are strongly enriched in potassium and resemble some types of shoshonitic magmas [63]. In this scenario, either the subducted crustal rocks mix with peridotites to form mélanges, or melts of subducted crustal rocks percolate and react with peridotites to form UP melts. This has previously been tested in experiments where a rhyolitic melt with as much as 6.4 wt% $K_2O$ was mixed with harzburgite [64], while others hybridized a phyllite (1.9 wt% $K_2O$) with dunite [30,31]. In both cases, experimental melts contain <6 wt% $K_2O$, and K/Na ratios of 4–5, much lower than the 12 wt% $K_2O$ and K/Na up to 10 of UP lavas [2]. It is noteworthy that crustal lithologies have a solidus up to 500 °C lower than that of peridotite [21]. Therefore, at mantle temperatures of >1200 °C, crustal lithologies generate Si-rich high degree melts in which $K_2O$ is diluted. This is evident from experiments hybridizing rhyolite and peridotite at 1200–1350 °C [19,64], where the maximum $K_2O$ mass fraction of the melt (~7.4 wt%) is close to that of the rhyolite component in the starting material (6.4 wt%). Even though these studies did not aim to reveal the origin of UP lavas, their work demonstrates that even high mass fractions of $K_2O$ in the starting material do not produce UP melts. In addition, K/Na ratios are lower than in the bulk starting material at >1100 °C, as melting of crustal rocks at high mantle temperatures does not form any Na-bearing phases that can increase K/Na ratios of the melt. All experimental melt compositions of sediment/peridotite hybridization so far fit this pattern [19,30,31,64], with K/Na values of 1–2 at ≥1100 °C (Figure 4A). In contrast, at <1100 °C, where temperatures are below that of the ambient mantle, experimental melts of either sedimentary rocks alone, or melts resulting from sediment/peridotite reaction, reach K/Na values that exceed those of the starting material. This difference in behavior is induced by the interaction between phengite stability and $Na_2O$ content of residual clinopyroxene in the sediment layer: as in the case of melting of phlogopite, the highest K/Na of experimental melts are reached after phengite breakdown (at 900 °C; Figure 4A) where all potassium is accommodated within the melt. However, at <1100 °C, the melts are generally Si-rich ($SiO_2$ > 70 wt%), thus high K/Na values are bound to high Si- and low-Mg contents, unlike UP lavas. These results are in accordance with previous experiments by the authors of [30,31], showing that melts produced by sediment + peridotite hybridization are mildly potassic and Si-rich. These may explain the origin of high-K calc-alkaline to shoshonitic lavas [7]. However, hybridization and melting of sediment and peridotite fail to explain the origin

of the most K-enriched lavas—lamproites—which form in the same areas as mildly potassic lavas [9].

## 5.2. First-Stage Experiments: Formation of Si-Rich Potassic Melts from a Phlogopite-Free Source

Melting of sedimentary rocks in the first-stage experiments and reaction of these melts with depleted peridotite produced Si-rich melts with K/Na >2, while total $K_2O$ never exceeded ~6 wt%, consistent with hybridization experiments in previous studies that used continental crust material [30,31]. The melt derived from the sediment is highly enriched in LILE, HFSE, and REE, and trace elements plot within the field of post-collisional lavas with K/Na >2 (Figure 5) [9]. It shows negative anomalies at Nb, P, and Ti, which are inherited from the bulk sediment composition. The formation of this Si-rich melt is explained by 40% batch melting of the bulk sediment composition (Supplementary Data Table S8):

$$100 \text{ wt\% sediment} \rightarrow 40 \text{ wt\% Si-rich melt} + 35 \text{ wt\% cpx} + 25 \text{ wt\% gt } (R^2 = 0.98). \tag{1}$$

The residue after melting of the sediment (clinopyroxene + garnet) of this first-stage process is rich in Na and Ca (Figure 3B,D). Clinopyroxenes from all first-stage reaction experiments are strongly zoned in Na across the reaction zone and reach highest $Na_2O$ mass fractions on the sediment side of the assemblage (Figure 2B). Additionally, Na is retained in clinopyroxene in the residue more effectively at a lower temperature (Figure 2B), corresponding to a decrease of $Na_2O$ partitioning from $D_{Na(cpx/melt)}$ ~2 at 3 GPa/800 °C to $D_{Na(cpx/melt)}$ ~0.8 at 3 GPa/1000 °C. The formation of residual Na-rich clinopyroxene and the breakdown of phengite produce melt with high K/Na ratios that peak at ~6.5 at 900 °C (Figure 4A). High K/Na ratios are also facilitated by the high Ca/Na composition of the sediment used in the experiments, which is probably typical for sediments during the late stages of ocean closure, where shallow basins above the calcite compensation depth facilitate the formation of carbonates. However, melts from the first-stage experiments at <1000 °C are $SiO_2$-rich (up to 79 wt%), which does not match the definition of UP magmas, but is common for melts from crustal lithologies [30]. Thus, UP melt generation requires a process that first increases the K/Na ratio of the melt at <1000 °C, which is facilitated by residual Na-rich clinopyroxene and high $D_{Na(cpx/melt)}$, and then reduction of the silica content by reaction with ultramafic material. However, at 900–1000 °C, hybridized melts are still Si-rich and show MgO contents <1 wt%, and hence do not satisfy the definition of ultrapotassic [30,31].

## 5.3. Second-Stage Experiments: Formation of High K/Na Melts from a Metasomatized, Phlogopite-Bearing Mantle

As hybridized melts of peridotite and crustal material are too low in MgO, $K_2O$, and K/Na, and too high in $SiO_2$, the formation of UP magma with MgO >3 wt%, K/Na >2, and $K_2O$ of 8 wt%–12 wt% from subducted crustal rocks must go beyond a single-stage mixing and melting process. Hence, a minimum of two stages is required: (1) Sediments or crustal rocks are subducted to mantle depth in a cool subduction setting where they melt, and the melt reacts with peridotite. At <1000 °C, the $SiO_2$-rich melts are consumed as they react with olivine to form phlogopite in an incongruent crystallization reaction (Figure 1A–D): melt + olivine → phlogopite + pyroxene. (2) During subduction and collision, phlogopite-bearing peridotite is separated from the K-poor residue, which resides within the foundering slab, while the K-enriched fore-arc is heated by rising asthenospheric mantle. The phlogopite pyroxenite melts at >1100 °C to produce UP melts with MgO >3 wt%, K/Na >2, and $K_2O$ of 8 wt%–12 wt%.

The phlogopite-bearing metasome forms in the first stage (Figure 3), and because the melt is silica-oversaturated, the reaction with olivine forms pyroxene in addition to phlogopite [24]. According to mass balance calculations (Supplementary Data Table S9), melt infiltration in the 3 GPa/900 °C experiment leads to the following reaction:

$$40 \text{ wt\% Si-rich melt} + 60 \text{ wt\% ol} \rightarrow 71 \text{ wt\% opx} + 21 \text{ wt\% phl} + 8 \text{ wt\% cpx } (R^2 = 0.997). \tag{2}$$

However, the spatial separation of phlogopite from orthopyroxene in the experimental reaction zones implies that this reaction occurs stepwise: the incongruent crystallization of phlogopite leads to the emergence of an $SiO_2$-rich fluid that migrates and reacts with olivine to cause the crystallization of orthopyroxene beyond the phlogopite-enriched layer. The initial silica-rich melt would produce orthopyroxene barriers that limit the width of the reaction zone [31], whereas fluid is much more mobile and infiltrates more quickly along olivine grain boundaries. This explains the observation that orthopyroxene veins protrude beyond the metasome into the dunite (Figure 1A–D; Supplementary Figure S2).

The UP melt of the second-stage experiment can be calculated to correspond to a ~20% batch melt of the metasome within the dunite (Supplement Data Table S10):

$$100 \text{ wt\% metasome (71 wt\% opx + 21 wt\% phl + 8 wt\% cpx)} \rightarrow 20 \text{ wt\% UP melt}$$
$$+ 56 \text{ wt\% opx + 19 wt\% ol + 5 wt\% phl (R}^2 = 0.999). \tag{3}$$

Reaction (3) produces UP melt and olivine by consuming phlogopite, clinopyroxene, and orthopyroxene of the phlogopite–pyroxenite metasome. The melting of phlogopite produces melts with potassium contents similar or higher than those of phlogopite in the source rock, as $K_2O$ is buffered by phlogopite at around 7 wt%–9 wt% $K_2O$ [16,60]. Incongruent melting of phlogopite produces olivine or orthopyroxene depending on the pressure [18], which itself causes an increase in $K_2O$ in the melt [65]. The trend of increasing K/Na with the degree of melting is also controlled by clinopyroxene as it changes its composition from Na-enriched to Na-depleted with increasing temperature [16]. As a result, the highest K/Na values in the melt are achieved at intermediate temperatures when phlogopite breaks down (Figure 4A). Hence, high K/Na ratios in UP magmas are inherited from their metasomatized source and reflect the relative proportions of clinopyroxene and phlogopite. The $SiO_2$ mass fractions reach values of ~57.7 wt% for the second-stage melts, which are in accordance with melting of a pure pyroxenite assemblage [28].

The dotted black line in Figure 4A shows the calculated K/Na of ~8.9 for the phlogopite pyroxenite that was estimated from the modal abundance of phlogopite and clinopyroxene within the reaction zone (Supplementary Data Table S2). As the metasome formed by the reaction of melt with olivine, which contains neither K nor Na, its K/Na is inherited entirely from the infiltrating melt, which reaches K/Na ~6.4. The difference between K/Na values of 6.4 and 8.9 may be a result of loss of clinopyroxene during transfer to the second-stage capsule. This transfer is depicted by the "metasomatized dunite" arrow in Figure 4A, which symbolizes the metasome formation and mechanical separation. Melting of the metasome produced UP melts with K/Na of ~7 and $K_2O$ mass fractions of ~8 wt%. While the K/Na ratio is similar to the first-stage silica-rich melt that metasomatized the dunite, the extreme enrichment of $K_2O$ during second-stage melting (3) was facilitated by physical separation from the K-poor residual sediment that formed by reaction (1). However, experimental melts are enriched in CaO by 2 wt%–3 wt% at a given mass fraction of $SiO_2$ (Figure 3D) compared with natural post-collisional lavas, which is likely because of initial Ca/Na of the Ca-rich sediment.

Owing to the small scale of the melt pools, trace elements could only be obtained from the 3 GPa/1000 °C experiment, which had melt pools beneath the reaction zone that were large enough for analysis by LA-ICP-MS. However, as the metasome formed by a reaction of 40 wt% melt of sediment with 60 wt% olivine (reaction 2), bulk trace element concentrations of the metasome can be calculated (Figure 5A, Supplementary Data Table S14). Also, trace element concentrations of the second-stage melt (reaction 3) can be modelled using the non-modal batch melting equation [66] with partition coefficients determined from partial melting of phlogopite pyroxenite [16]. While the bulk metasome composition shows concentrations below those observed in post-collisional lavas [9], 10–30% batch melts of it lie within the observed compositional range (Figure 5A).

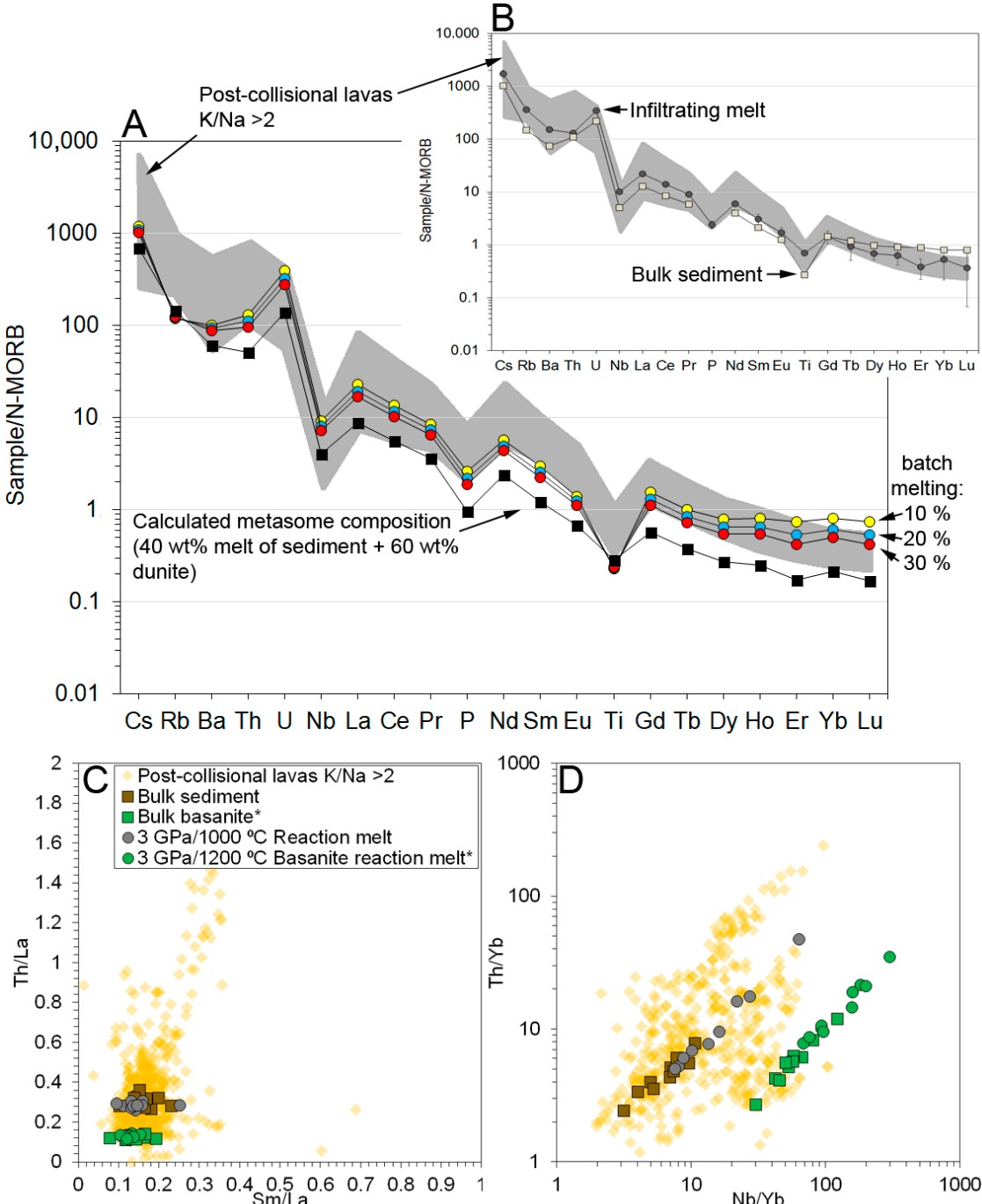

**Figure 5.** N-MORB normalized trace element compositions of (**A**) modelled batch melt compositions and (**B**) starting materials [67] (grey squares, sediment) compared with experimental melts (grey circles, 3 GPa/1000 °C) and post-collisional lavas (shaded area) [9]. The sediment/dunite reaction melts (B) are rich in Cs, Rb, and Ba, and show negative anomalies at Nb, P, and Ti, similar to post-collisional lavas with K/Na > 2. (A) Forty weight percent melt of sediment reacts with 60 wt% dunite to form the calculated metasome composition. Batch melting of the metasome (10–30%) increases the trace element mass fractions to values observed within post-collisional lavas. (**C**) Th/La versus Sm/La shows experimental 3 GPa/1000 °C sediment/dunite reaction melts at Th/La 0.2–0.6, plotting with the bulk of post-collisional lavas [9], while peridotite melts such as basanite/dunite reaction melts plot at Th/La <0.2 [53]. (**D**) Th/Yb versus Nb/Yb of sediment/dunite reaction melts plot with post-collisional lavas, while basanite/dunite reaction melts attain higher Nb/Yb ratios and plot largely outside the field of post-collisional lavas [53].

Post-collisional lavas show a strong enrichment of Th over REE. Reaction melts in the 3 GPa/1000 °C experiment plot in Th/La versus Sm/La and Th/Yb versus Nb/Yb within the field of post-collisional lavas [9], which attain high Th/La and Th/Yb (Figure 5C,D). In contrast, melts of peridotite and basanite/dunite reaction melts [53] show low Th/La and Th/Yb. The composition of the 3 GPa/1000 °C sediment/dunite reaction melt scatter around the sediment composition at Th/La ~0.3 and Sm/La 0.1–0.2 and suggest that heterogeneities within the source components lead to variations within post-collisional lavas (Figure 5C). Th/Yb versus Nb/Yb are positively correlated in experimental samples and post-collisional lavas (Figure 5D), whereby melts with no crustal input have the lowest Th/Yb and high Nb/Yb, as evident from the basanite/dunite reaction experiments [53]. Nevertheless, Th/La ratios of lamproites reach high values of >1 (Figure 5C) that are yet unmatched by experimental melt compositions [20], suggesting either an enigmatic process for Th/La fractionation or another subducted protolith.

Our results demonstrate a general process where subducted sediments lead to mantle metasomatism and K-enrichment. It is very likely that different sediment compositions and varying degrees of melting of the sediment will lead to less, or even more, K-enrichment of the fore-arc mantle. On balance, we demonstrate that Mediterranean sediment leads to metasomatism and the formation of K-rich melts similar to high-Si lamproites found in the Mediterranean area [28].

*5.4. Implications for the Formation of Post-Collisional Lavas*

The sediment/dunite reaction of the first-stage experiments produced melts with 3 wt%–6 wt% $K_2O$ and $SiO_2$ varying from 55 wt%–79 wt% with decreasing temperature from 1000 to 800 °C. Their major element (Figures 2 and 3) and trace element concentrations (Figure 5B) are in accordance with the formation of post-collisional lavas by model (B) that are Si-rich and show $K_2O$ concentrations similar to shoshonites and high-K calc-alkaline lavas [9]. Although melt inclusions in garnet indicate that ultrahigh-pressure melting of paragneiss may produce melts with high Th/U and LILE as well as fractionating LREE from HREE, the K-enrichment by this process is insufficient to produce lamproites; melts are more similar to some types of shoshonite [63] or granites [68]. None of the first-stage experiments of this study produced K-rich melts that are comparable to lamproites, which coexist with shoshonites and high-K calc-alkaline lavas, nor have other experiments that investigated sediment–peridotite hybridization [30,31]. The most K-enriched crustal rocks, rhyolites and granites, will also produce only mildly potassic lavas if they hybridize with peridotites [19,64]. Hence, the direct melting of crustal lithologies and hybridization with mantle rocks proposed in model (B) is only able to explain the formation of mildly potassic, Si-rich, post-collisional lavas.

However, it must be emphasized that post-collisional lavas of shoshonitic and high-K calc-alkaline composition occur concurrently in time and space with extremely K-enriched lamproites in several volcanic provinces of the AHOB [28,29]. In two well-documented examples, extremely K-enriched lavas were erupted in a fore-arc setting, induced by slab-rollback, and indicate a strongly metasomatized lithospheric fore-arc mantle [33,34]. Accordingly, the simultaneous appearance of lamproites with other less K-rich lavas suggests that they share a similar source and represent an end-member with the highest proportion of metasomatized mantle (phlogopite pyroxenite) in their source. Melts of phlogopite pyroxenites, as demonstrated by the second-stage melting experiment, produce melts with a high $K_2O$ mass fraction of 8 wt%–9 wt% and high K/Na of ~7, which are comparable to post-collisional lavas with lamproitic compositions (Figures 2 and 4), in accordance with previous studies on phlogopite pyroxenite melting [16]. Batch-melting modelling of the second-stage experiments shows that 20% melting of phlogopite pyroxenite satisfies major and trace element composition of high-K post-collisional lavas (Figures 2 and 5). The high degree of melting (20%) contrasts with the extremely low-degree melts (<2%) that are required to form high-K alkaline magmas from a peridotitic source (e.g., [5]) and better explains high volumes of K-rich lavas observed in areas such as the Roman Province (e.g., [69]). However, the overall scarcity of extremely ultrapotassic lavas such as lamproites may be explained by the rarity of K-rich mantle metasomes.

## 6. Conclusions

The re-use of the sediment-melt metasomatized dunite confirms that two-stage formation can account for UP magmatism with $K_2O$ >6 wt% involving significant contributions from silicic crustal components, as seen in almost all post-collisional UP magmas. Extremely K-enriched magmas require a modally metasomatized mantle enriched in phlogopite [8,9,11,34]. This enrichment probably takes place during subduction when sediment starts to melt at ~675 °C, forming high K/Na dacitic/rhyolitic melts that react with wall-rock peridotite. Phlogopite is formed by incongruent crystallization during this reaction, consuming the melt (Figure 6A). The residue of the melted sediment attains a low K/Na ratio and crystallizes Na–Al rich clinopyroxene, releasing melt with high K/Na. After metasomatism and enrichment of the fore-arc in K/Na, the low K/Na residue is subducted to greater depths and temperatures where it may contribute to arc magmas with K/Na <1 (Figure 6A).

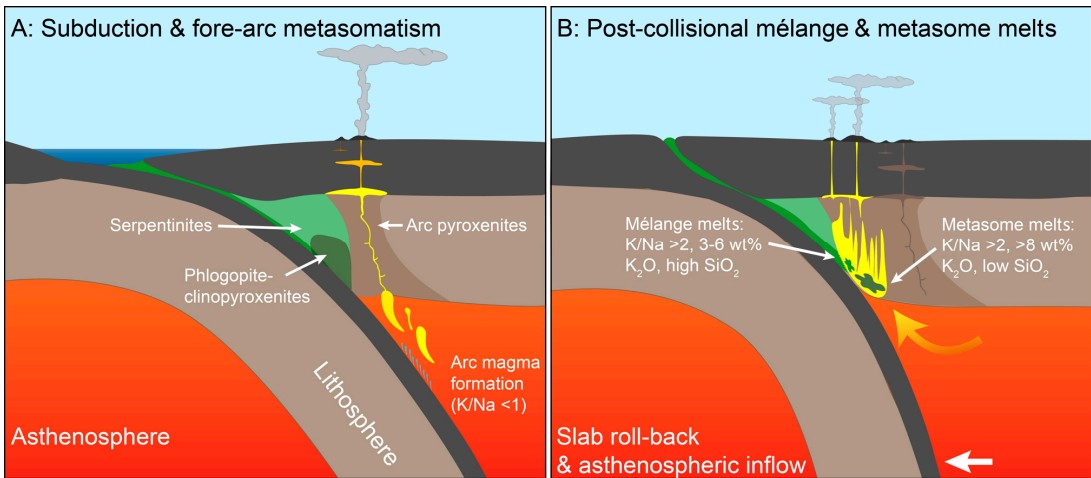

**Figure 6.** Model for the genesis of post-collisional UP magmatism. (**A**) Fore-arc mantle is metasomatized by melts of subducting sedimentary rocks; melts have K/Na ratios >2 and react with depleted mantle to form phlogopite clinopyroxenites, leaving a K-depleted residuum that contributes at deeper levels to arc magmas with low K/Na ratios of <1. (**B**) After slab-rollback (and possibly during continental collision), upwelling asthenosphere introduces heat and melts the metasomes and mélanges. These second-stage magmas are ultrapotassic in composition (K/Na > 2, $K_2O$ 3 wt%–6 wt% from crust, and mélanges and up to >8 wt% for metasome melts).

After subduction ceases, slab-rollback and/or break-off allows access of hot asthenospheric mantle to shallow levels of the fore-arc mantle, thereby heating the modally metasomatized peridotite (Figure 6B). Partial melting of the phlogopite-enriched metasome now produces UP melts with $K_2O$ contents of 6 wt% to 12 wt%. In addition, large amounts of subducted crustal lithologies incorporated into newly-formed lithosphere [28] may directly melt to produce siliceous potassic rocks with lower $K_2O$ (<6 wt%; Figure 6).

However, if the rise of the hot asthenospheric mantle fails to induce melting of the metasomes, they may reside within the lithospheric mantle until they are "activated" at a much later stage. This later activation could be induced by geodynamic processes different to the post-collisional setting, including heat introduction by mantle plumes, or rifting and associated decompression melting. Potassic magmatism that exhibits the involvement of subducted sediments remote to young orogens in locations such as Gaussberg; Antarctica [70,71]; or West Kimberley, Australia [72,73] would fit this scenario.

**Supplementary Materials:** The following are available online at http://www.mdpi.com/2075-163X/10/1/41/s1, Figure S1: Residual sediment of the 3 GPa/900 °C experiment containing ~60% clinopyroxene (grey), ~40% garnet (light grey), and interstitial glass (dark grey) as estimated by Fiji (imagej)., Figure S2: Orthopyroxene-veines in

dunite of the 3 GPa/900 °C experiment containing 22% orthopyroxene (grey) and 78% olivine (red) as estimated by Fiji (imagej). The metasomatized sediment is located below the phlogopite layer, Table S1: Glass averages. Table S2: Starting materials, Table S3: 3 GPa/800 C Reaction Experiment. Table S4: 3 GPa/850 C Reaction Experiment, Table S5: 3 GPa/900 C Reaction Experiment, Table S6: 3 GPa/1000 C Reaction Experiment. Table S7: 2nd-stage melting experiment 2 GPa/1200 C, Table S8: Mass balance of 3 GPa/900 C sediment melt, Table S9: Mass balance of 3 GPa/900 C reaction zone, Table S10: Mass balance of 2nd-stage experiment using the metasomatized dunite as starting composition, Table S11: Reference materials for LA-ICP-MS quality control, Table S12: Solution ICP-MS measurements, Table S13: Trace elements of 3 GPa/1000 C sediment-dunite reaction experiment, Table S14: Trace element modelling.

**Author Contributions:** Conceptualization, M.W.F., D.P., B.X., and S.F.F.; Methodology, M.W.F. and S.B.; Formal Analysis, M.W.F., S.B. and R.M.-K.; Writing—Original Draft Preparation, M.W.F., D.P., B.X., and S.F.F.; All authors have read and agreed to the published version of the manuscript.

**Funding:** This work is part of the lead author's (M.W.F.) PhD thesis supported by an Australian Government International Postgraduate Research Scholarship (IPRS) and Postgraduate Research Fund (PGRF). D.P. was supported through the Deutsche Forschungsgemeinschaft (DFG) project PR 1072/9-1. S.F.F. is funded by ARC grant FL 180100134.

**Acknowledgments:** We gratefully acknowledge Qing Xiong for providing the dunite sample ZD11–53 used in the experiments. The International Ocean Discovery Project (IODP) supported this study in providing the Mediterranean marine sediment sample from sampling site 161-976. This is contribution 1417 from the ARC Centre of Excellence for Core to Crust Fluid Systems (http://www.ccfs.mq.edu.au) and 1360 in the GEMOC Key Centre (http://www.gemoc.mq.edu.au).

**Conflicts of Interest:** The authors declare no conflict of interest.

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
