# Peer review of "Two-Stage Origin of K-Enrichment in Ultrapotassic Magmatism Simulated by Melting of Experimentally Metasomatized Mantle"

_minerals, doi:10.3390/min10010041_

Round 1

Reviewer 1 Report

The reviewed article focuses on petrological problem of ultrapotassic magmatism mainly in a post-collisional setting.

The article focusing on generation of highly potassic (up to 12 wt% K2O) melts in the mantle provides results of two-stage experimental study of formation of ultrapotassic magmas and discusses the range of melting and melt-rock reaction processes that take place on the different depth levels within mantle wedge.

The authors used a novel two-stage experimental approach in which parts of an experimental product are re-used in a second experiment.

On the base of the two-stage experiments the authors stresses that (i) the highly potassic and mildly-low siliceous magmas (lamproite-type) are  generated by melting of a modally metasomatized mantle source that has been enriched in phlogopite, (ii)  the phlogopite-enriched metasome is produced by reaction of depleted peridotite with K-Si-rich melts (high-K calc-alkaline type) that has been generated by melting of sedimentary rocks.

I have read the article with great interest, as it addresses wide‐ranging topics of modern petrology (phlogopite in the mantle, contribution of subducted sediments to a post-collisional and, more intriguing, to an intraplate magmatism, melt-rock reactions within the lithospheric mantle and others). ..

The topic of article is adequate to modern petrological trend to study mantle volatile-bearing minerals as phlogopite and amphybole. Recent publications (e.g. Saha & Dasgupta, 2019; Safonov et al., 2019) support this opinion. Novelty of the article is related with explanation that K-rich melts required for phlogopite metasomes occur, have been generated from subducted sedimentary source during a separate event.

Undoubtedly, the topic of article will be of interest to the wide-range readers of the journal.

The composition of the article is clear and logical. The Introduction is problem-oriented. The conclusions are supported by the data.  References are relevant but near 30% of them are referenced to authors’ studies.

The methods described with sufficient details in the Section 3.2 and Supplementary-data-tables-11-14

I find the paper well written and of great interest to scientists in this field.

Despite the overall good impression of the article, some recommendations for improving can be made.

It would be preferably to give examples of phlogopite-rich pyroxenite that has been documented in both cratonic and island arc xenoliths. The review (Safonov et al., 2019) and study (Bryant et al., 2007; Aulbach et al., 2017; Kargin et al., 2017) will be usefull. It is necessary to give more attention to lamproites when described problems of the article in the Introduction. The lamproites mentioned only once in the introduction  and have been detailed discussed in 4.1, 4.3, 4.4 sections. It would be usefully for readers to stress similarities and differences between orogenic and cratonic lamproites considering the idea of the authors that phlogopite-rich metasome may reside within the lithospheric mantle until they are “activated” at a much later by by mantle plumes, or rifting.

 Minor comments

The article has high-level presentation style and minimum minor comments are required.

The text lines 428-438 is diffuse and poorly understood in comparison with a simple and clear Figures 5 C and 5D. Please, shorten and simplify the text.

Despite the explanation in lines 360-369, the suggested mechanism for the spatial separation of phlogopite from orthopyroxene more evidence of it realize in the nature requires.

Saha, S., & Dasgupta, R. (2019). Phase relations of a depleted peridotite fluxed by a CO2‐H2O fluid—Implications for the stability of partial melts versus volatile‐bearing mineral phases in the Cratonic Mantle. Journal of Geophysical Research: Solid Earth, 124. https://doi. org/10.1029/2019JB017653

Safonov O., Butvina V., Limanov E. (2019). Phlogopite-Forming Reactions as Indicators of Metasomatism in the Lithospheric Mantle. Minerals, 9, 685; doi:10.3390/min9110685

Bryant, J. A., G. M. Yogodzinski, and T. G. Churikova (2007), Melt-mantle interactions beneath the Kamchatka arc: Evidencefrom ultramafic xenoliths from Shiveluch volcano,Geochem. Geophys. Geosyst.,8, Q04007, doi:10.1029/2006GC001443.

Aulbach S., Sun J., Tappe S., Höfer H. E., Gerdes A., (2017). Volatile-rich Metasomatism in the Cratonic Mantle beneath SW Greenland: Link to Kimberlites and Mid-lithospheric Discontinuities, Journal of Petrology, Volume 58, Issue 12,  Pages 2311–2338, https://doi.org/10.1093/petrology/egy009

Kargin A.V., Sazonova L.V., Nosova A.A., Lebedeva N.M., Tretyachenko V.V., Abersteiner A., Cr-rich clinopyroxene megacrysts from the Grib kimberlite, Arkhangelsk province, Russia: Relation to clinopyroxene–phlogopite xenoliths and evidence for mantle metasomatism by kimberlite melts, Lithos, Volumes 292–293, 2017, Pages 34-48, ISSN 0024-4937, https://doi.org/10.1016/j.lithos.2017.08.018.

Author Response

It would be preferably to give examples of phlogopite-rich pyroxenite that has been documented in both cratonic and island arc xenoliths.

We incorporate additional literature now.

The review (Safonov et al., 2019) and study (Bryant et al., 2007; Aulbach et al., 2017; Kargin et al., 2017) will be usefull. It is necessary to give more attention to lamproites when described problems of the article in the Introduction. The lamproites mentioned only once in the introduction and have been detailed discussed in 4.1, 4.3, 4.4 sections. It would be usefully for readers to stress similarities and differences between orogenic and cratonic lamproites considering the idea of the authors that phlogopite-rich metasome may reside within the lithospheric mantle until they are “activated” at a much later by by mantle plumes, or rifting.

We added the additional literature and extended the introduction.

The text lines 428-438 is diffuse and poorly understood in comparison with a simple and clear Figures 5 C and 5D. Please, shorten and simplify the text.

We shortened and simplified this part.

Despite the explanation in lines 360-369, the suggested mechanism for the spatial separation of phlogopite from orthopyroxene more evidence of it realize in the nature requires.

We extended this section accordingly.

Reviewer 2 Report

Review of the paper of Michael W. Förster et al. “Two-stage origin of K-enrichment in ultrapotassic magmatism simulated by a novel experimental approach”

The manuscript presents an experimental study on the formation of ultrapotassic melts through melting of mantle peridotite metasomatically transformed to phlogopite-bearing pyroxenite under the influence sediment-derived potassium-rich silicic melt. The experimental model includes two steps, one of which illustrates the formation of phlogopite and pyroxenes in depleted dunite. These experiments are interesting in that they expand our knowledge on peridotite - sediment interaction in subduction-related mélange zones. The second stage simulates melt production in the metasomatized peridotite enriched in phlogopite and showing a high K/Na ratio. Its partial melting results in the formation of ultrapotassic hydrous melts similar in composition to natural ultrapotassic melts, including lamproites. The model is reasonable, but some points should be discussed in more detail.

First, the authors must demonstrate the advantages of their approach, involving the use of materials from the first-stage interaction experiments, in the second-stage partial melting experiment. The products of the interaction experiments are heterogeneous and consist of phlogopite, pyroxenite, and olivine-bearing layers. The proportions of these materials used in the partial melting experiment are arbitrary and may be not applicable to natural situations. A more common approach would involve mixing of minerals whose composition was determined in the interaction experiment and melting such a new synthetic mixture with a more realistic (and controlled) bulk composition. Similar experiments were referred to by the authors [14-17 in the paper], and it was noted that they produced ultrapotassic liquids. Such experiments allow estimation of the influence of bulk metasomatized peridotite composition on melting relations. This is important, because some features of the composition of partial melt are controlled by mineral equilibria, whereas others are dependent on the bulk composition of the source and degree of melting (incompatible elements). The behavior of K is uncertain, because its content can be only partly buffered by melt saturation with olivine and phlogopite. Obviously, Al activity should also be an important controlling factor.

Second, phlogopite is not the only possible source of K for the formation of ultrapotassic magmas. Under water-deficient conditions, other K hosts in peridotite could be K-Mg carbonate and anhydrous K-bearing silicates (e.g. Brey et al., 2011, Chem. Geol., 281, 333-342).

Third, the authors should be more cautious in their inferences, keeping in mind that they rely on the results of a single experiment. The choice of experimental conditions for this experiment should be justified to show that it is applicable to natural environments. There is no data to assess the influence of other factors (P-T conditions, presence of other fluid components, proportions of materials in the melting zone among others) on the composition of the produced melt. I think that the results are very interesting and important for petrologists, but further experimental efforts are needed to obtain reliable quantitative estimates.

            There are some comments on certain places in the text.

Lines 49-51. The models of ultrapotassic rock generation are more diverse and include also extensive crystal fractionation and crustal assimilation, which were previously reviewed by one of the authors (Foley et al., 1987).

Line 83. What is the novelty of the experimental approach? The use of experimental products in later experiments is a rather common experimental technique (e.g. Baker and Stolper, 1994, GCA, 58, 2811-2827; Widmark, 1980, CMP 72, 175-179; and many others). Lundstrom (2003, Geochem. Geophys. Geosyst. 4(9) 8614) used a two-step method for studying partial melting and element transport. A search of the experimental literature will provide many other similar experiments.

Line 134. Oxygen fugacity is not controlled in the experiments by C+CO (more correctly, C+CO+CO2) equilibrium, but is definitely lower because of the complex composition of the C-O-H fluid.

Lines 174-178. Considerations on gas phase composition are unclear. First, why the “decomposition of the sediment follows the reaction: CH4 + H2O = 3 H2 + CO” if sediments contain neither CH4 nor free H2O? Second, where is the quantity 6.9 wt % O from? Third, “given carbon saturation” and presence of melt and residual mica, the amounts of gases in the fluid phase cannot be calculated from the bulk contents of the elements. In my opinion, this paragraph should be omitted.

Lines 194-200. It must be mentioned that all component percentages in melts are recalculated to 100 wt % anhydrous totals (as in Table 2).

Lines 219-220. What accessory minerals? The meaning of this sentence is unclear.

Line 222 and further. Try to be consistent with rock names. “Metasomatized dunite” must be dominated by olivine, and strongly metasomatized material should be referred to as phlogopite pyroxenite after dunite.

Line 233. KD(ol/liq) decreasing, not increasing.

Line 254. 2.46 wt % K2O in Table 2.

Line 353. “Phlogopite + pyroxene” must be in the right-hand side of the reaction (as in reaction 2).

Line 354. What is the mechanism of separation of phlogopitized peridotite from K-poor residue?

Lines 367-369. It is highly probable that pressure and temperature gradients in the experimental samples are much higher than those in “a dynamic subduction setting.”

Line 384. Change “K/Na content” to K/Na.

Lines 486-487. In fact, the amount of lamproite-like magmas in magmatic provinces is always small compared with basaltic magmatism. Hence, the large degree of melting of the metasomatized mantle seems to disagree with the natural observations. The relative rarity of metasomatized mantle domains can be invoked to reconcile this discrepancy.

Table 2. Please, give glass analyses not normalized to 100%, or provide analytical totals. Why is the total of the starting sediment 101%?

Table 2, 3. What is “first standard deviation”? Maybe, one standard deviation?

Author Response

Lines 49-51. The models of ultrapotassic rock generation are more diverse and include also extensive crystal fractionation and crustal assimilation, which were previously reviewed by one of the authors (Foley et al., 1987).

Agreed, in this case we only want to compare the models of K-enrichment within Earth’s mantle and not processes that follow upon magma fractionation. We modified the sentence accordingly.

Line 83. What is the novelty of the experimental approach? The use of experimental products in later experiments is a rather common experimental technique (e.g. Baker and Stolper, 1994, GCA, 58, 2811-2827; Widmark, 1980, CMP 72, 175-179; and many others). Lundstrom (2003, Geochem. Geophys. Geosyst. 4(9) 8614) used a two-step method for studying partial melting and element transport. A search of the experimental literature will provide many other similar experiments.

Apologies, we were not aware that the same material has been used again (e.g. Lundstrom 2003) and had always the impression that second stages were produced by synthetical mixing an experimental composition that represents the analysed run products of the first stage. We modified title and text accordingly.

Line 134. Oxygen fugacity is not controlled in the experiments by C+CO (more correctly, C+CO+CO2) equilibrium, but is definitely lower because of the complex composition of the C-O-H fluid.

Yes, but it limits the fO2 to being not higher than the CCO equilibrium. Re-worded this part.

Lines 174-178. Considerations on gas phase composition are unclear. First, why the “decomposition of the sediment follows the reaction: CH4 + H2O = 3 H2 + CO” if sediments contain neither CH4 nor free H2O? Second, where is the quantity 6.9 wt % O from? Third, “given carbon saturation” and presence of melt and residual mica, the amounts of gases in the fluid phase cannot be calculated from the bulk contents of the elements. In my opinion, this paragraph should be omitted.

The CH4 and H2O would be generated by the breakdown of the organic material. We agree that this is very speculative and omit this part.

Lines 194-200. It must be mentioned that all component percentages in melts are recalculated to 100 wt % anhydrous totals (as in Table 2).

Added

Lines 219-220. What accessory minerals? The meaning of this sentence is unclear.

We removed this sentence as it is not necessary here.

Line 222 and further. Try to be consistent with rock names. “Metasomatized dunite” must be dominated by olivine, and strongly metasomatized material should be referred to as phlogopite pyroxenite after dunite.

Changed throughout the manuscript.

Line 233. KD(ol/liq) decreasing, not increasing.

Corrected

Line 254. 2.46 wt % K2O in Table 2.

Corrected

Line 353. “Phlogopite + pyroxene” must be in the right-hand side of the reaction (as in reaction 2).

Corrected

Line 354. What is the mechanism of separation of phlogopitized peridotite from K-poor residue?

Subduction, the Na-enriched part stays within the sinking slab. Re-worded this part.

Lines 367-369. It is highly probable that pressure and temperature gradients in the experimental samples are much higher than those in “a dynamic subduction setting.”

We agree and therefore removed this sentence since the comparison of both systems is to speculative.

Line 384. Change “K/Na content” to K/Na.

Corrected

Lines 486-487. In fact, the amount of lamproite-like magmas in magmatic provinces is always small compared with basaltic magmatism. Hence, the large degree of melting of the metasomatized mantle seems to disagree with the natural observations. The relative rarity of metasomatized mantle domains can be invoked to reconcile this discrepancy.

We agree and added a sentence accordingly.

Table 2. Please, give glass analyses not normalized to 100%, or provide analytical totals. Why is the total of the starting sediment 101%?

We now present analytical totals. The starting sediment of 101% was a typo and has been corrected to 100%.

Table 2, 3. What is “first standard deviation”? Maybe, one standard deviation?

Corrected

Reviewer 3 Report

Review of “Two-stage origin of K-enrichment in ultrapotassic magmatism simulated by a novel experimental approach” by Förster et al., submitted to Minerals

Dear editor, dear authors

This paper reports on high-pressure, high-temperature experiments that aim to investigate the enrichment of potassium in ultrapotassic magmas. After the presentation of the two petrological models used to explain this particular type of magmatism (melting of a phlogopite-rich metasome vs. partial melting of subducted sediments and/or continental crust), the authors design their experiments in order to test the metasome melting model. A two-layer charge made of dunite (to simulate the mantle) and marine sediment (to simulate the crust) produce a clinopyroxene+phlogopite reaction zone that is subsequently extracted from the charge and remelted in a second experimental run. The authors succeed in obtaining K-rich melts with geochemical signatures similar to natural ultrapotassic lavas. The results of this research are subsequently considered in the large context of the evolution of active plate margins, where post-collisional ultrapotassic magmas occur.

Melt-rock interaction in subduction/collisional settings is undoubtedly a scientific problem of interest for the petrologic community. I think that the manuscript would make a worthwhile contribution to Minerals, at least for this reviewer point of view, and I commend the authors for their experimental design. Some moderate additional work is needed to improve the submitted version. Here are a few comments the authors could consider:

Give more emphasis to the novelty of the study. This research confirms what we already know on the generation of ultrapotassic magmas, i.e. the melting of metasomes rich in phlogopite. The authors should dig more into the available literature and find what is still unknown about the melting metasome model. For example, do the composition of the fluid released by the crust into the mantle have a role in the generation of ultrapotassic magmas? Is the CO2 component of the crustal fluid important to stabilize, or not, clinopyroxene in the reaction zone? The authors have in hand a fluid composition (lines 176-178) that they do not really exploit during the rest of the manuscript. The interpretation of the microstructure of the reaction zone produced during the first melting stage is not convincing. I see a double corona, with clinopyroxene+mica towards the crust and an olivine+orthopyroxene layer on the mantle side. The veins are not really visible, judging from the pictures provided. In addition, figure 1D shows an almost monomineralic opx shell between phlogopite and olivine. This mineral arrangement reminds me of corona textures related to diffusion of elements along chemical potentials gradients. I suggest the authors to reconsider the interpretation of this microstructure or, at least, provide compelling evidence of the opx+ol veins. Line 187; reaction 1; line 353. I keep wondering the reason why silica-rich melts reacting with olivine does not produce orthopyroxene but, instead, clinopyroxene. Is this related to the COH fluid composition? Line 354. It is unclear why the phlogopite-bearing peridotite should separate from the residue. Is it a gravity-driven process? Or a mechanical tearing? And why this separated portion should reach the hottest part of the subduction zone? Please, clarify. Lines 388-389 vs line 397. K/Na and Ca/Na ratios are explained with loss and excess clinopyroxene, respectively, during the transfer of the reaction zone from the first to the second melting stage. This is not clear, at least to me. Please, clarify. Line 462. An interesting reference on SiO2-rich melt inclusions in garnet from orogenic peridotite is the one of Alessia Borghini, Geology 2018. Line 493. Sediment melting at 800°C requires nearly fluid-absent conditions, while in this research the presence of COH fluids should significantly lower the crust solidus. Please, reconsider this conclusion. Finally, are there field examples of metasomes similar to those reproduced in the crucible? The authors have to sustain their inferred process with example from crust-mantle mélange around the world or from volcanic centers erupting crustal and ultramafic xenoliths. This would strengthen the interpretation of the authors and increase the impact of their results.

I hope the above comments will help the authors clarifying their manuscript.

Author Response

Give more emphasis to the novelty of the study. This research confirms what we already know on the generation of ultrapotassic magmas, i.e. the melting of metasomes rich in phlogopite. The authors should dig more into the available literature and find what is still unknown about the melting metasome model. For example, do the composition of the fluid released by the crust into the mantle have a role in the generation of ultrapotassic magmas?

We extended the introduction and discussion accordingly. However, we cannot give more information on the fluid compositions and reduced this part as reviewer 2 suggested.

Is the CO2 component of the crustal fluid important to stabilize, or not, clinopyroxene in the reaction zone? The authors have in hand a fluid composition (lines 176-178) that they do not really exploit during the rest of the manuscript.

The formation and stabilization of cpx is provided by the Ca-content of the sediment melt. We present this in part 5.3.

The interpretation of the microstructure of the reaction zone produced during the first melting stage is not convincing. I see a double corona, with clinopyroxene+mica towards the crust and an olivine+orthopyroxene layer on the mantle side.

We agree that phlogopite and clinopyroxene form a corona on the sediment side of the reaction zone. However the dunite side is more complex and also contains opx veins (see supplement). We modified this part accordingly.

The veins are not really visible, judging from the pictures provided. In addition, figure 1D shows an almost monomineralic opx shell between phlogopite and olivine. This mineral arrangement reminds me of corona textures related to diffusion of elements along chemical potentials gradients.

We show within supplementary figure 2 that opx veins are found along the grain boundaries to olivine. We colored the Ol grains with imagej in red and opx veins (grey) are easily recognizable.

I suggest the authors to reconsider the interpretation of this microstructure or, at least, provide compelling evidence of the opx+ol veins. Line 187; reaction 1; line 353. I keep wondering the reason why silica-rich melts reacting with olivine does not produce orthopyroxene but, instead, clinopyroxene. Is this related to the COH fluid composition?

We provide evidence in the supplement that opx veins are present within the dunite part of the experimental charge. The formation of cpx instead of opx in the reaction zone is due to the Ca-content of the sediment melt as shown in 5.3.

Line 354. It is unclear why the phlogopite-bearing peridotite should separate from the residue. Is it a gravity-driven process? Or a mechanical tearing? And why this separated portion should reach the hottest part of the subduction zone? Please, clarify.

The process of Na separation from K is facilitated by normal subduction. The Na-enriched part stays within the sinking slab (in eclogites). Re-worded this part.

Lines 388-389 vs line 397. K/Na and Ca/Na ratios are explained with loss and excess clinopyroxene, respectively, during the transfer of the reaction zone from the first to the second melting stage. This is not clear, at least to me. Please, clarify.

Yes this is actually a contradiction. The higher Ca/Na of the melts thus must be from high Ca/Na of the starting sediment. We changed this accordingly.

Line 462. An interesting reference on SiO2-rich melt inclusions in garnet from orogenic peridotite is the one of Alessia Borghini, Geology 2018.

Added

Line 493. Sediment melting at 800°C requires nearly fluid-absent conditions, while in this research the presence of COH fluids should significantly lower the crust solidus. Please, reconsider this conclusion.

Yes, we already present in the introduction that sediments melt at T as low as 675 C. Changed this part accordingly.

Finally, are there field examples of metasomes similar to those reproduced in the crucible? The authors have to sustain their inferred process with example from crust-mantle mélange around the world or from volcanic centers erupting crustal and ultramafic xenoliths. This would strengthen the interpretation of the authors and increase the impact of their results.

We have integrated new literature that reports on mantle xenoliths with compositions similar to the metasomes in our reaction runs.